# Octacosanol Modifies Obesity, Expression Profile and Inflammation Response of Hepatic Tissues in High-Fat Diet Mice

**DOI:** 10.3390/foods11111606

**Published:** 2022-05-30

**Authors:** Jie Bai, Tao Yang, Yaping Zhou, Wei Xu, Shuai Han, Tianyi Guo, Lingfeng Zhu, Dandan Qin, Yi Luo, Zuomin Hu, Xiaoqi Wu, Feijun Luo, Bo Liu, Qinlu Lin

**Affiliations:** 1Hunan Key Laboratory of Grain-Oil Deep Process and Quality Control, College of Food Science and Engineering, Central South University of Forestry and Technology, Changsha 410004, China; t20071464@csuft.edu.cn (J.B.); yangtao807@163.com (T.Y.); zyp4265@163.com (Y.Z.); wxu537@163.com (W.X.); 20190100076@csuft.edu.cn (S.H.); guotianyib11@163.com (T.G.); zhulingfeng1988@163.com (L.Z.); 20200100080@csuft.edu.cn (D.Q.); huzuomin97100214@163.com (Z.H.); a19118999340@163.com (X.W.); leerychen@outlook.com (B.L.); t20081475@csuft.edu.cn (Q.L.); 2Hunan Key Laboratory of Forestry Edible Resources Safety and Processing, National Research Center of Rice Deep Processing and Byproducts, Changsha 410004, China; 3Department of Clinic Medicine, Xiangya School of Medicine, Central South University, Changsha 410008, China; yiluo603@hotmail.com

**Keywords:** octacosanol, hyperlipidemia, gene chip, NF-κB, lipid metabolism, AMPK

## Abstract

The incidence of obesity has increased significantly on account of the alterations of living habits, especially changes in eating habits. In this study, we investigated the effect of octacosanol on lipid lowering and its molecular mechanism. High-fat diet (HFD)-induced obesity mouse model was used in the study. Thirty C57BL/6J mice were divided into control, HFD, and HFD+Oct groups randomly, and every group included ten mice. The mice of HFD+Oct group were intragastrically administrated 100 mg/kg/day of octacosanol. After 10 weeks for treatment, our results indicated that octacosanol supplementation decreased the body, liver, and adipose tissues weight of HFD mice; levels of TC, TG, and LDL-c were reduced in the plasma of HFD mice; and level of HDL-c were increased. H&E staining indicated that octacosanol supplementation reduces the size of fat droplets of hepatic tissues and adipose cells comparing with the HFD group. Gene chip analysis found that octacosanol regulated 72 genes involved in lipid metabolism in the tissues of liver comparing to the HFD group. IPA pathway network analysis indicated that PPAR and AMPK may play a pivotal role in the lipid-lowering function of octacosanol. Real-time quantitative PCR and Western blot showed that the octacosanol supplementation caused change of expression levels of AMPK, PPARs, FASN, ACC, SREBP-1c, and SIRT1, which were closely related to lipid metabolism. Taken together, our results suggest that octacosanol supplementation exerts a lipid-decreasing effect in the HFD-fed mice through modulating the lipid metabolism-related signal pathway.

## Highlights:

Octacosanol exerts a lipid-decreasing effect in the HFD-fed mice.

Octacosanol inhibits inflammation via MAPK/NF-κB signaling in the liver tissues.

Octacosanol changes the gene expression profile of liver tissues.

Octacosanol modulates the lipid metabolism-related signal pathways.

## 1. Introduction

The incidence of obesity has increased significantly on account of the alterations of living habits, especially changes in eating habits. Hyperlipidemia is vital factor of obesity, characterized by the abnormal increase of one or more lipid components, mainly referring to triglycerides (TG) and total cholesterol (TC) in plasma, and is a kind of metabolic disorder syndrome [1]. Hyperlipidemia as a risk factor for cardiovascular disease. If not treated in time, patients will suffer from serious diseases, such as coronary heart disease, myocardial infarction, and stroke. Therefore, controlling blood lipid concentration is the key way to prevent people from cardiovascular and cerebrovascular diseases [2]. At present, the drugs used in the clinic are statin, bette, bile-acid-chelating agents, and nicotinic acid, and these drugs exert lowering-lipid effects in different ways. However, some patients have a series of adverse reactions, such as rhabdomyolysis, liver damage, and mental depression [3,4]. It is of great clinical significance to search for a type of new lipid-lowering drugs with long-lasting effects, mild lipid-lowering effects, and low-level toxicity. Investigators found that curcumin has no or less toxicity and can be used as a potential candidate for the treatment of hyperlipidemia [5]. Dietary starfish oil also prevented live steatosis and hyperlipidemia in C57BL/6J mice induced by high-fat diet [6]. Comparing with drug therapy of hyperlipidemia, natural phytochemicals provide new clues for the treatment of hyperlipidemia, which provide patients superiority in safety and tolerance.

Octacosanol is the main component of policosanol [7]. When the content is up to 12.5%, policosanol shows a significant lipid-lowering activity [8,9]. Octacosanol is a natural high-fat alcohol and widely found in plants, animals, and their secretion and is especially rich in plant lipids, such as rice, wheat, and other cereals [8,9]. Nowadays, the preparation of octacosanol is still mainly from cane wax, rice bran wax, and beeswax, and rice bran wax is the best source. The total output of rice bran is about 30 million tons per year globally. This makes it possible to explore the physiological function of octacosanol in depth.

It has been proven that policosanol and octacosanol could inhibit the accumulation of fat, affect lipid metabolism, and promote the metabolism of cholesterol [10]. The effect on lowering blood lipids was related to dose by comparing the high and low doses of octacosanol and gradually reduced with the prolongation of administration time [10]. But the underlying lipid-lowering mechanism of octacosanol is still unclear. We still do not know what genes and what signal pathways are involved in the effect. In this study, we used high-fat diet mice to evaluate lipid-lowering effects and used gene chip to analyze the alterations of gene expression profile after octacosanol treatments, bioinformatics, and molecular experiments to explore the possible target genes and signal pathways.

## 2. Materials and Methods

### 2.1. Feed Formula of Mice

The basic diet of mice contained 38.0% wheat, 20.0% maize, 18.0% soybean meal, 10.0% fish meal, 5.0% wheat bran, 3.0% soybean oil, 2.0% maltodextrin, 1.0% minerals, and 1.0% vitamins (SLAC, Changsha, Hunan, China). The HFD mice feed consisted of 79.6% basic diet supplemented with 1.0% cholesterol, 0.2% sodium bile acid, 0.2% propylthiouracil, 10.0% lard, 5.0% egg yolk powder, and 4.0% milk powder. The feed formula of control and HFD mice was described in our published paper [11].

All animal experiments involved in this study were reviewed and approved by the Guidelines for the Care and Use of Experimental Animals, the Hunan Normal University, College of Medicine (SYXK-Xiang, 2015-0007). All mice were obtained from air-conditioned animal rooms under specific pathogen-free and 12/12 h reverse light/dark cycles conditions, and free access to water and food. Wild-type C57BL/6J male mice of 7 to 8 weeks old were individually housed and maintained at an environment of 23 ± 2 °C and adapted to the experimental condition based on stable food and water intake.

### 2.2. Octacosanol Treatment

All mice were randomly divided into three groups (10 per group) at random after an adaptation period of one week: the control (Con), HFD, and HFD+Octacosanol (HFD+Oct) group. The Con group was given basic diet and water ad libitum, while the HFD or HFD+Oct group were given a high-fat diet. Octacosanol was additionally administrated to mice of HFD+Oct group by gavage (200 mg/Kg) 3 days before HFD feeding until the end of the experiment about 11 weeks later. As a parallel measurement, the other two groups were intragastric administrated the same volume of water simultaneously every day.

### 2.3. Glucose Tolerance Testing

After fasting for 6 h, the glucose tolerance of mice was performed by intraperitoneal (i.p.) injection of 2 g of 25% glucose (Sigma-Aldrich, St. Louis, MO, USA). The blood glucose concentrations in the tail of mice were measured by a glucometer (Hemocue, Angelholm, Sweden) at the time points of 0, 15, 30, 60, and 120 min. Data are described as mean concentration of blood glucose per group using area under the curve (AUC; glucose challenge).

### 2.4. Analysis of Blood Biochemical Indicators

The whole blood samples of mice were obtained from anterior cubital veins by venipuncture, adding heparin and ethylenediaminetetraacetic acid (EDTA) immediately, and swiftly centrifuging to separate plasma. The plasma parameters of TC, TG, high-density lipoprotein (HDL), and low-density lipoprotein (LDL) were measured. The assay method refers to our previous publication [12].

### 2.5. Histological Analysis

The whole liver and retroperitoneal adipose tissues were carefully removed after sacrifice of mice. About 100 mg liver and retroperitoneal adipose tissues from the same site were intercepted for histological analysis, fixed in 10% (100 g/L) formalin solution for over 24 h, and dehydrated in absolute ethanol. Then the tissues were transplanted with xylene before paraffin imbedding, cut into 5 μm sections with microtome, mounted on clean glass slides, and finally dried overnight at 37 °C. After dewaxing with xylene and dehydrating with absolute ethanol, prepared tissue sections were subjected to hematoxylin and eosin (H&E) staining, followed by observing under microscope [12].

### 2.6. RNA Isolation and Array Processing

Liver tissues were lysed using Transzol Up (TransGen, Beijing, China) under liquid nitrogen cooling to extract the total RNA. The obtained RNA samples were stored in −80 °C refrigerator. NanoDrop ND-2000 spectrophotometer (Thermo Fisher Scientific Company, Waltham, MA, USA) and standard denaturing agarose gel electrophoresis were used to evaluate RNA concentration quality and integrity, respectively. Using Invitrogen SuperScript ds-cDNA synthesis kit, 5 μg of total RNA was synthesized into double-stranded cDNA (ds-cDNA), which next were washed and labeled according to the NimbleGen Gene Expression Analysis Protocol (NimbleGen Systems, Inc., Madison, WI, USA): 4 μg of RNase was incubated with ds-cDNA for 10 min at 37 °C and cleaned with chloroform: isopropyl alcohol, then precipitated with absolute ethanol at 4 °C. The purified cDNA was quantitatively analyzed using NanoDrop ND-2000. According to the manufacturer’s guidelines in the Gene Expression Analysis protocol (NimbleGen Systems, Inc., Madison, WI, USA), the cDNA was labeled with Cy3 using NimbleGen One-Color DNA Labeling Kit. One OD Cy3 primer was incubated with 1 μg of ds-cDNA for 10 min at 98 °C, followed by addition of 100 pmol of deoxynucleoside triphosphates and 100 U of the Klenow enzyme (New England Biolabs, Ipswich, MA, USA), and incubation at 37 °C for 2 h. Thereafter, the reaction was terminated by adding 0.1 vol of 0.5 M EDTA, and the purified ds-cDNA was precipitated with isopropanol/ethanol. In a hybridization chamber (Hybridization System-NimbleGen Systems, Madison, WI, USA), the microarray was subjected to hybridization with 4 μg of Cy3-labeled ds-cDNA in the hybridization buffer/hybridization fraction A at 42 °C for 16 h. After carefully washing with Wash Buffer (NimbleGen Systems, Madison, WI, USA), the gene chips were scanned (Axon GenePix 4000B microarray scanner) after washing in an ozone-free environment. Differentially expressed genes between the two groups were identified by fold-change filtration or by *t*-test filtration. Hierarchical clustering was performed using MeV software (4.4v). Protein molecular network obtained from octacosanol-modified genes of upregulated and downregulated expressions were analyzed by IPA software.

### 2.7. RT-PCR Analyzing mRNA Expressions

The CFX96 Real-Time PCR system (Applied Biosystems Co., Ltd., Foster City, CA, USA) was used for real time quantitative RT-qPCR analysis performed by SYBR^®^ Select Master Mix Kit (Applied Biosystems Co., Ltd.). According to the manufacturer’s protocol (Applied Biosystems Co., Ltd., Foster City, CA, USA), expression levels of genes involved in proinflammatory cytokines (IL-6 and TNF-α), lipid metabolism (SIRT1, PPAR, ACC, and CD36), and β-actin (control) were analyzed. The relative expression level of mRNA was measured by PCR system (Applied Biosystems, USA) software represented by the ratio of expression level of target gene to endogenous gene showed in bar graph. The PCR primers used in the experiments was described in our previous publication [12].

### 2.8. Western Blot Analysis

The prepared protein samples (10−20 μg) extracted from mice liver tissues using 400 μL RIPA buffer supplemented with 1% cocktail, 1% phosphatase inhibitors, and 1% phenylmethylsulfonyl fluoride (Roche, Basel, Switzerland) were mixed well with an equal volume of 5× sodium dodecyl sulfate (SDS) (125 mM Tris-HCl (pH = 6.8), 4.0% SDS, 10.0% 2-mercaptoethanol, 0.3% bromophenol blue, and 20.0% glycerol) and then boiled for 10 min at 95 °C. The mixture was subjected to 10–15% SDS-polyacrylamide gel electrophoresis (PAGE) gel electrophoresis; then, the proteins contained in corresponding gel were taken out according to the molecular weight of the target gene and were transferred onto polyvinylidene difluoride (PVDF) membrane. Next, 5.0% bovine serum albumin (BSA) (2 g BSA in 40 mL 1× Tris-buffered saline with Tween 20 (TBS-T)) was blocked and the membrane combined with protein of target gene at 25 °C, which was then incubated for overnight with a mixture of primary antibody diluted 1:1000 with 5% BSA in TBST at 4 °C. The incubated PVDF membrane was washed 3 times with TBST for 30 min at room temperature. After incubated for 1 h at 4 °C with an anti-mouse/anti-rabbit IgG secondary antibody diluted 1:5000–1:10,000 and washed with TBST for 30 min again, and immunoreactive proteins on membrane were detected using an ECL Plus Western blotting (WB) Detection System (Pierce, Rockford, IL, USA) according to the manufacturer’s protocol and imaged in a Gel Imaging System (Chemi-Doc XRS^+^, Bio-Rad). The relative expression level of the target key protein and control was obtained by calculating the integrated optical density of each band using Gel Imaging System. Anti-β-actin (#58169), Anti-FASN#3180, Anti-ACC (#4190), Anti-CD36 (#14347), Anti-PPAR-α (#74076), Anti-P-PPAR-γ (#8660TSer82) Anti-PPAR-γ (#2435), Anti-LXR (#61723), Anti-AMPK (#2532), and Anti-P-AMPK (#8208 Thr172) were purchased from Cell Signaling Technology, Danvers, CO, USA; Anti-SREBP-1c (sc-8984), Anti-SREBP-1c (sc-8984), and Anti-SRIT1 (sc-15404) were purchased from Santa Cruz Biotechnology, USA. Anti-P-PPAR-α (Ab3484 Ser12) was purchased from Abcam Biotechnology, Cambridge, UK; Anti-P-PPAR-δ (#PA5-105082Thr256) was purchased from Thermo Fisher Scientific, Sunnyvale, CA, USA.

### 2.9. Statistical Analysis

Each experiment was repeated three times. Statistical analyses were performed using SPSS Statistics software (Version 2.50, SPSS Inc., Chicago, IL, USA). The differences among groups were analyzed using one-way analysis of variance test. Data shown as the mean ± standard deviation (SD) were considered statistically significant at *p*-value less than 0.05. Correlations between two variables were determined by Pearson’s correlation coefficient.

## 3. Results

### 3.1. Octacosanol Enhances Glucose Tolerance of HFD Mice

Glucose tolerance refers to the ability of the body to regulate blood glucose concentration. After eating high-starch foods, such as rice and flour staple foods, or taking glucose, the glucose is absorbed by the intestine, makes the body’s blood sugar rise, and stimulates the insulin secretion, and the liver glycogen synthesis increases, decomposition is suppressed, the liver glycogen output reduces, and after, the body of blood sugar is maintained in a relatively stable range. After 11 weeks’ supplement of octacosanol in mouse experiments, the results showed that the area under the glucose tolerance curve (AUC) increased by 23.74%, and the AUC decreased by 17.44% in mice fed with octacosanol compared with normal control mice; the results showed that octacosanol could enhance the glucose tolerance of obese mice, as shown in Figure 1.

### 3.2. Octacosanol Ameliorate the Obesity Phenotype of HFD-Induced Mice

The mice fed with octacosanol for 11 weeks had a slight decrease in food intake, but there was no statistically significant difference. The weight gain of mice in the Oct+HFD group was significantly lower than that in HFD group but still higher than that in the Con group. The average body weight of HFD mice raised from 23.50 ± 0.91 to 26.72 ± 2.86 g after 11 weeks of HFD (*p* < 0.01). The average weight of mice treated with octacosanol was 25.35 ± 1.93 g (*p* < 0.05). As shown in Figure 2A, the weight gain of the mice in HFD group was significantly higher than that in Con group, while octacosanol reduced the weight gain. These results suggest that octacosanol partially inhibited the weight gain in the HFD mice.

The liver of mice in HFD group was larger, whiter, smoother, and greasier than those in the normal group, and octacosanol significantly inhibited the liver enlargement and whitening induced by HFD, as shown in Figure 2B. Compared with Con group, the liver weight in the HFD group raised from 1.49 ± 0.15 g to 1.65 ± 0.27 g (*p* < 0.01). Octacosanol supplementation significantly improved liver damage induced by HFD, with liver weight reduced to 1.52 ± 0.13 g, as shown in Figure 2B. These data show that octacosanol can alleviate liver weight gain in hyperlipidemic mice and liver fat accumulation induced by HFD.

At the end of the experiment, the weight of the retroperitoneal fat pad was measured. HFD diet resulted in more significant abdominal fat deposits than the Con group, while octacosanol observably reduced the weight and size of the retroperitoneal fat tissue, as shown in Figure 2. The weight of adipose tissue in Con group, HFD group, and HFD+Oct group were 1.46 ± 0.09, 1.84 ± 0.17, and 1.56 ± 0.07 g, respectively, as shown in Figure 2. The above evidence shows that octacosanol can effectively inhibit the HFD-induced abdominal fat deposition in mice.

### 3.3. Histological Alterations of Liver Tissues and Epididymal Fat

To estimate the effect of octacosanol on the histology and morphology of mice, livers and epididymal fat were cut into slices and stained by H&E. The image of liver pathological sections of mice in each group (Figure 3A) showed that the hepatocyte in Con group were normal in morphology and distribution, and the cytoplasm was homogeneous without fatty degeneration, while hepatocyte of mice in HFD group were swollen and arranged as loosely scattered with the white vacuole formed by different lipid droplets. The hepatocyte of mice in HFD+Oct group was near the normal hepatocyte in Con group, and the white vacuole was less than HFD group. Compared with HFD group, octacosanol treatment can reduce the epididymal fat cell hypertrophy induced by high-fat diet (Figure 3B). All these data indicate that octacosanol may lower the lipids of liver by reducing the lipid accumulation and prevent the formation of fatty liver to a certain extent.

### 3.4. Octacosanol Improves the Plasma Lipid Level in High-Fat Diet Mice

The blood samples of mice were collected to acquire the lipid profiles of plasma. Compared with Con group, TG level was obviously increased in the HFD group mice; the TG level in HFD+Oct group was 45.96% (*p* < 0.05, Figure 4A) lower than that in HFD group. The results indicate that octacosanol could obviously reduce TG level of plasma in HFD mice. Likewise, compared with Con group, TC level was increased in the HFD group mice; TC level in HFD+Oct group was significantly reduced by 26.93% compared with HFD group (*p* < 0.05, Figure 4B), indicating that intervention with octacosanol can significantly inhibit the increase of TC level in HFD mice. Compared with HFD group, LDL-c level in HFD+Oct group was decreased by 47.93% (*p* < 0.05, Figure 4C). Compared with HFD group, HDL-c level in HFD+Oct group decreased by 11.22% (*p* = 0.06, Figure 4D), and the difference was not significant (*p* ˃ 0.05, Figure 4D).

### 3.5. Octacosanol Modulated the Gene Expression Profile of Liver Tissues

cDNA microarray was applied to analyze the gene expression in mouse liver tissues. There were 72 differentially expressed genes in related to lipid, which was initially confirmed by PubMed database retrieval among more than 15,000 genes, including 44 downregulated genes (Table 1) and 28 upregulated genes (Table 2). By combining with hierarchical clustering analysis, it can be seen from the heatmap that there are 72 obviously different expression profiles between the two groups (Figure 5). Comparing with Con group, 44 genes were upregulated expressions in the liver tissues and octacosanol supplementation inhibited the 44 gene expressions. Comparing with Con group, 28 genes were downregulated expressions in the liver tissues, and octacosanol supplementation upregulated the 44 gene expressions (Figure 4).

### 3.6. GO and KEGG Analyzed the 72 Different Expression Genes

The genes were divided into 18 categories according to their biological functions, including fatty acid metabolic process, response to muscle stretch, metabolism of lipids, lipid homeostasis, regulation of lipid metabolic process, insulin resistance, and so on. About half of the alternated genes are closely associated with lipid metabolism (see Figure 6). KEGG analysis indicated that octacosanol supplementation modulated PPAR pathway, sphingolipid pathway, fatty acid degradation pathway, insulin resistance pathway, and lectin receptor pathway, and all of those pathways are closely related with lipid metabolism. Meanwhile, octacosanol supplementation also regulated the Toll-like receptor pathway, which is a key pathway of inflammation, suggesting octacosanol may inhibit HFD-induced inflammation through the Toll-like receptor pathway (see Figure 7).

### 3.7. IPA Analysis Indicated That Octacosanol Modulated AMPK and ERK Pathways

To explore the lipid-lowering underlying mechanism of octacosanol, the 72 differentially expressed genes were further analyzed by signal pathway analysis and the construction of protein network via IPA software. After octacosanol supplementation, the differentially expressed genes affecting the lipid-lowering function were significantly enriched in the ERK pathway and AMPK signaling pathway (Figure 8). These data showed that the two kinases ERK and AMPK might play an important role in the octacosanol lipid-lowering function.

### 3.8. Octacosanol Affected Lipid Metabolism-Related Gene Expressions

To further reveal the possible mechanism of the lipid-lowering function of octacosanol, the mRNA levels and protein levels of lipid metabolism-related genes were examined by RT-qPCR analysis and WB test, respectively. The data of RT-qPCR showed that the mRNA expressions of those genes, such as PPAR, SIRT-1, ACC, and SREBP-1c, were evidently increased in HFD mice compared with the Con group (Figure 9A). WB data indicated that the protein expressions of those genes, such as PPAR, SIRT-1, and ACC, were also significantly increased in HFD mice compared with the Con group (*p* < 0.05) (Figure 9B). However, the protein levels of SREBP-1c did not change between HFD group and HFD+Oct group. It suggests that octacosanol may exert lipid-lowering function through regulating the expression of genes related to lipid metabolism.

### 3.9. Octacosanol Affected PPAR and AMPK Signal Pathways

To further reveal the possible signal pathways of the lipid-lowering effect of octacosanol, protein samples of liver tissues were analyzed by WB. The test results showed that HDF promoted phosphorylations of PPAR-α (p-PPAR-α) and p-PPAR-γ but did not affect p-PPAR-δ, suggesting that HDF can activate PPAR signal pathway. Octacosanal supplementation could decrease HDF-induced phosphorylations of PPAR-α and PPAR-γ, which means octacosanol can inhibit PPAR signal pathway (Figure 10A). Compared with the HFD group, octacosanol promoted phosphorylation of AMPK (pAMPK-T172) in the liver tissue of mice and significantly increased the protein level of pAMPK (Figure 10B). These showed that octacosanol may inhibit lipid metabolism-related gene expressions by modulating the AMPK signal pathway and PPAR signal pathway.

### 3.10. Octacosanol Decreases Inflammatory Factor Expressions

Considering that the inflammatory reaction is related to the increase of blood lipids [13], the expression levels of inflammatory cytokines were analyzed by RT-qPCR and WB. The experiment results showed that HFD can increase the expressions of TNF-α, IL-6, and iNOS and octacosanol can remarkably reduce the expression levels of these factors by RT-qPCR, *p* < 0.01, and *p* < 0.05, respectively (Figure 11). WB analysis confirmed that octacosanol could inhibit the protein expression of these inflammatory cytokines (Figure 11). It suggests that octacosanol can restrain the expression levels of inflammatory cytokines in RNA and protein layers, thus reducing the inflammation accompanied with obesity caused by a high-fat diet.

## 4. Discussions

Unhealthy diets and lacking exercise result in a growing population of people with obesity. Obesity patients are usually characterized by a significant increase of triglycerides and total cholesterol. Now, most of lipid-lowering drugs used in clinical treatments have different side effect in human bodies. It is necessary to explore novel natural compounds to exert the lipid-lowering effect and decrease the side effect of lipid-lowering drugs. Previous investigations showed that some foods or natural compounds, such as polysaccharides, flavonoids, and glycosides, could reduce plasma lipid levels without obvious side effects [14,15,16]. In this study, our research found that octacosanol supplementation in diets significantly reduced HFD-induced obesity in mice. Octacosanol reduced the weight gain in mouse bodies, lipid accumulation in the liver, and epididymal tissues in the HFD-induced mice. Plasma lipid profiles of TC, TG, and LDL-c was decreased, and that of HDL-c was upregulated, meaning octacosanol could significantly ameliorate blood lipid index of HFD mice. The result is consistent with recent reports [10].

To explain lowering-lipid mechanisms of octacosanol, gene chips analysis was used to analyze the gene expression profiles of hepatic tissues. Our study is the first investigation to report expression profile alternated by octacosanol supplementation in HFD mice, which benefit us to comprehensively understand the lowering-lipid effect of octacosanol. The alternated expression genes are related with lipid metabolism, including the changed expressions of lipid metabolism-related genes and multiple signal pathways of lipid metabolism. Further IPA analysis also confirms the discovery and constructed the protein network modulating by octacosanol supplementation.

In this study, our data showed that the classic lipid metabolism-related genes, such as SREBP1, ACC, PPARα, and FASN, are modulated by octacosanol, and it also regulated AMPK pathway in the liver tissues of HFD mice. The kinase AMPK is the vital regulator of energy metabolism, which could reduce ATP-consuming processes and upregulate ATP regeneration [12,17]. AMPK takes part in many biological functions: the pathological processes of obesity, hyperlipidemia, and diabetes. The AMPK has different types of heterotrimeric complexes, which include an alpha-catalytic, a beta-regulatory, and a gamma-regulatory subunit. Threonine phosphorylation of AMPK is necessary to activate the kinase, and activated AMPK promotes the oxidation of fatty acids and also reduces the synthesis of cholesterol, fat, and TG, which causes the decrease of lipid accumulation. On the contrary, AMPK inactivation could reduce the oxidation of fatty acids and cause hyperlipidemia [16,17,18]. In the study, we found that octacosanol supplementation promoted AMPK phosphorylation and activation, which means the lipid-lowering effect of octacosanol may be mediated by AMPK pathway.

SIRT1 is an important metabolism and energy sensor and belongs to NAD+-dependent deacetylase. SIRT1 usually takes part in homeostatic responses to nutrient availability, and SIRT1 is the key regulator of lipid metabolism-related gene expressions [19,20]. In the study, our results indicated that octacosanol supplementation promoted SIRT1 expression in the liver tissues of HFD mice. In fact, SIRT1 can activate AMPK and decrease Fas expression, which can reduce lipid accumulation in liver tissues. Interestingly, AMPK can also activate SIRT1 and trigger deacetylation by upregulating NAD+ levels, which modulates lipid metabolism-related gene expressions and exerts a lipid-lowering effect.

Many investigations showed that AMPK activation, SREBP-1, and ACC inhibition could ameliorate hyperlipemia therapy. AMPK can phosphorylate ACC and promote ACC inactivation. ACC can catalyze malonyl-CoA production from acetyl-CoA, and it is an important rate-limiting enzyme of fat synthesis. ACC plays a vital role in the fatty acids metabolism, and ACC activation could modulate fatty acid synthesis [21,22]. SREBP is an important lipogenic transcription factor and controls lipogenesis, which includes the synthesis of fatty acids and triglycerides. AMPK can prevent SREBP-1 activation, thereby decreasing the transcriptional expressions of fatty acid and TG synthesis-related enzymes [23]. In the study, HFD decreased AMPK activation and promoted SREBP1-mediated lipogenesis. Meanwhile, FAS, an SREBP1 target protein, upregulated expression. The FAS gene is fatty acid synthase and directly participates in fatty acid synthesis, which causes an accumulation of TC and TG [24]. AMPK can prevent ACC activation and FASN expression, which decreases lipid accumulation, suggesting that octacosanol supplementation exerts a lipid-lowering effect that may be mediated by the AMPK/SREBP1/ACC pathway.

Plasma cholesterol content can be modulated by transcription factors SREBPs and LXRs. LXRs are the most abundantly expressed in the liver and control multiple important processes of cholesterol metabolism [25,26]. For example, LXRs can prevent cholesterol absorption in the gut and increase its reverse transport, which causes the convert of bile acid and then promotes bile acid excretion [25]. In the study, LXR expression was obviously reduced in the liver tissues of HFD mice, which causes the imbalance of cholesterol metabolism and leads to hyperlipidemia. On the contrary, octacosanol increased LXR expression in the liver tissues of HFD mice, suggesting that the octacosanol may modulate the LXR pathway and regulate cholesterol metabolism.

PPARs belong to the ligand-activated nuclear receptor superfamily, and they have three subtypes (PPARα, PPARγ, and PPARδ). PPARs have many different biological functions, including carcinogenesis, inflammation, and lipids metabolism [27,28,29]. PPARs especially modulate many gene expressions of lipids metabolism. PPARα is highly expressed in liver tissues and has high mitochondrial and â-oxidation activity, which participates in lipid metabolism; PPARδ regulates gene expressions of lipid oxidation and energy dissipation; PPARγ is a transcription factor and takes part in adipocyte differentiation and fat formation. Activated PPARα activation can promote fatty acid oxidation and reduces the level of circulating triglyceride; meanwhile, it also can reduce lipid storage. AMPK could modulate expressions of PPARα/PPARγ and their target genes [27,30]. In this study, our results indicated that octacosanol can activate AMPK and partly reduce the expressions of PPARγ and PPARδ; meanwhile, octacosanol increased the PPARα expression in the HFD mouse model, suggesting that PPARs are also important target and participates in the anti-hyperlipidemic function of octacosanol.

Previous investigations indicate that obesity and high-fat symptoms can cause chronic inflammation in the body. Excessive fat accumulation can promote MAPK-NF-κB activation, which promotes the expressions and secretions of pro-inflammatory factors [11]. In the study, our results indicated that HFD also increased the expressions of pro-inflammatory factors in mouse liver tissues, and octacosanol supplementation decreased the expressions of pro-inflammatory factors. MAPK is the important signal pathway to regulate the expressions of inflammatory factor [31,32]. MAPK can activate NF-κB, and it is the vital transcript factor and controls the transcript expressions of many inflammatory factors, and the promoters of most of inflammatory factors include one or more binding sites of NF-κB [33,34,35]. Octacosanol supplementation could inhibit NF-κB activation, suggesting that octacosanol supplementation prevents inflammation and may be mediated by regulating NF-κB in the HFD mice. This result matches our previous investigation that octacosanol inhibits inflammation through the MAPK/NF-κB pathway.

In conclusion, our investigation showed that octacosanol could ameliorate hyperlipidemia in the HFD-induced mouse model, and the lipid-lowering function was mediated by multiple pathways, including AMPK/SREBP-1c/ACC, AMPK/SIRT1, PPARs, and LXR pathways. Octacosanol could reduce ameliorate inflammatory response in the liver tissues of HFD-induced mice, and the anti-inflammation effect was mediated by MAPK/NF-κB pathway. Taken together, dietary interventions using natural compounds such as octacosanol might be a candidate for an adjuvant therapy for hyperlipidemia.

## Figures and Tables

**Figure 1 foods-11-01606-f001:**
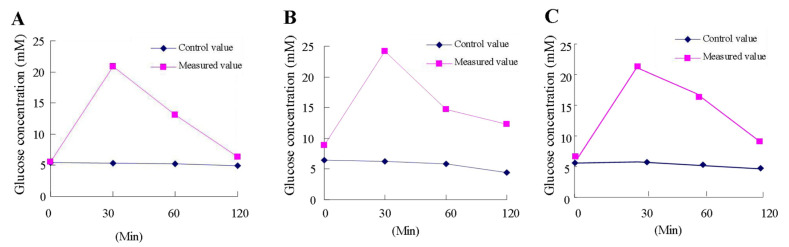
Octacosanol could enhance the glucose tolerance in the high-fat diet mice. (**A**) Con group; (**B**) HFD group; and (**C**) HFD+Oct group.

**Figure 2 foods-11-01606-f002:**
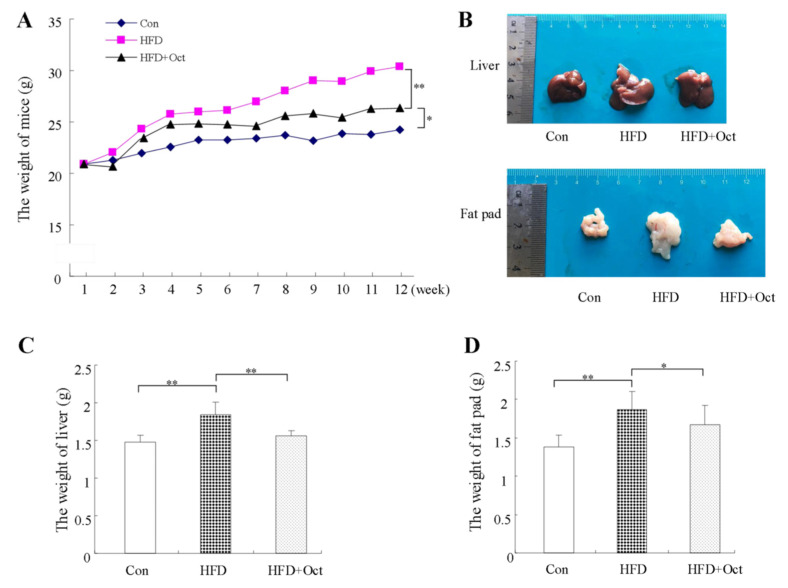
Effects of octacosanol on HFD-induced mice bodyweight gain. Thirty 8-week-old C57BL/6J mice were randomized into three groups (10 per group), and the mice from the Control, HFD, and HFD+Oct group were fed for 10 weeks. (**A**) Bodyweight change of mice during 10 weeks. (**B**) Representative image of liver and fat pad of mice at the end of feeding; (**C**) effect of octacosanol on the live weight at the end of feeding; (**D**) effect of octacosanol on the weight of retroperitoneal fat pads at the end of feeding. Data presented as mean ± SD, *n* = 10. Oct, octacosanol; HFD, high-fat diet; **: *p* < 0.01; *: *p* < 0.05.

**Figure 3 foods-11-01606-f003:**
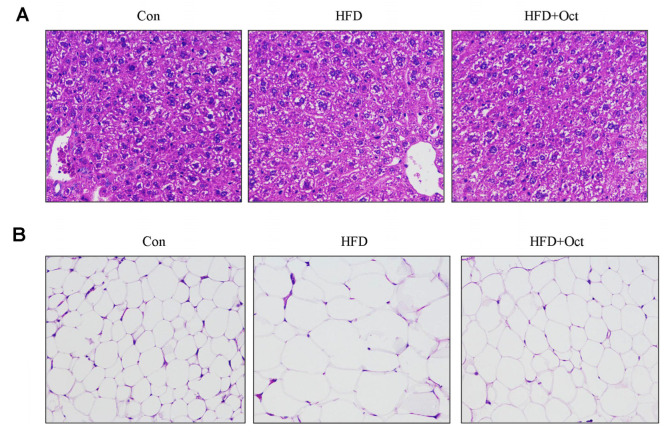
Octacosanol alleviates HFD-induced hepatic damage and decrease adipocytes size of retroperitoneal fat tissue. (**A**) H&E staining images of representative hepatic tissues of Con, HFD, and HFD+Oct groups; (**B**) H&E staining images of representative retroperitoneal fat tissue of Con, HFD, and HFD+Oct groups. Oct, octacosanol; HFD, high-fat diet.

**Figure 4 foods-11-01606-f004:**
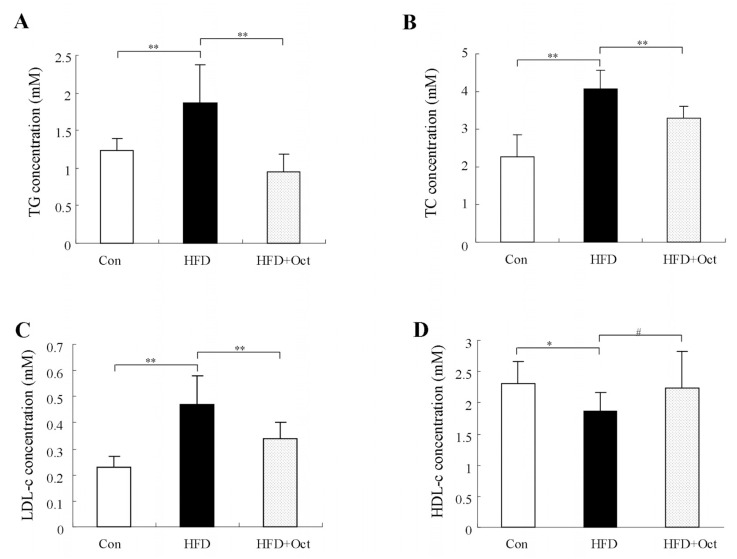
Octacosanol attenuates abnormal lipid levels in the plasma of high-fat diet mice. (**A**) Concentrations of triglycerides (TG) from different groups; (**B**) concentrations of total cholesterol (TC); (**C**) concentrations of high-density lipoprotein (HDL); (**D**) concentrations of low-density lipoprotein (LDL). Data presented as mean ± SD, n = 10. Oct, octacosanol; HFD, high-fat diet; **: *p* < 0.01; *: *p* < 0.05; ^#^: *p* > 0.05. Octacosanol attenuates abnormal lipid levels in the plasma of high-fat diet mice.

**Figure 5 foods-11-01606-f005:**
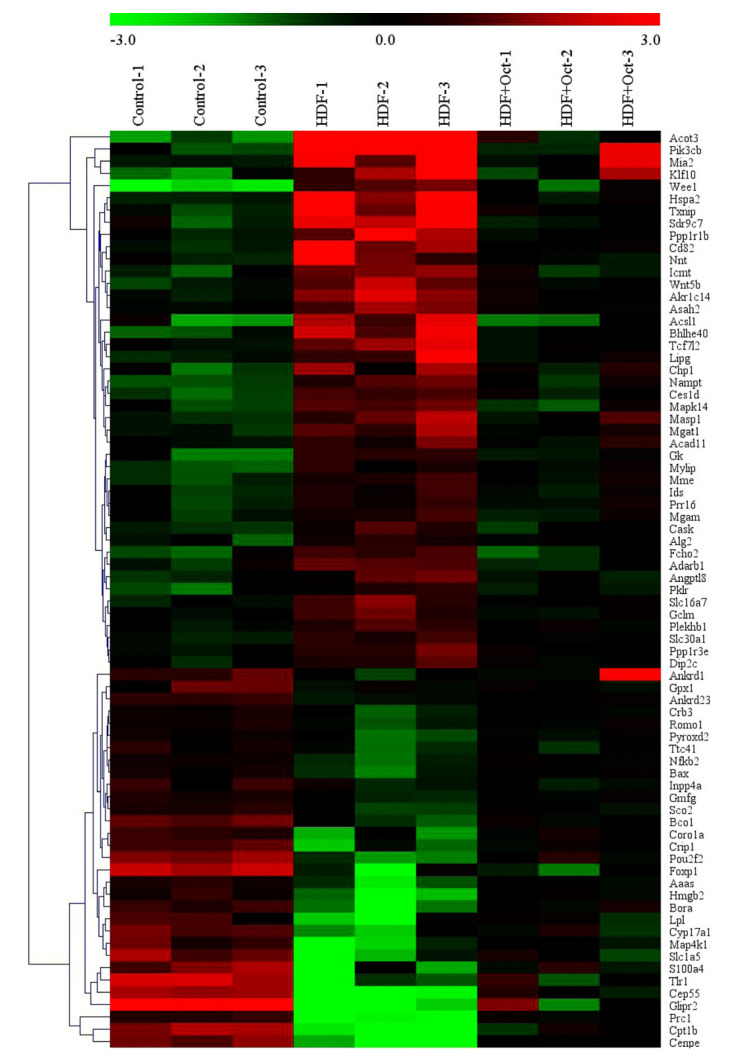
Clustering analysis showed the changed genes of HFD and HFD+Oct groups comparing with control group. Seventy-two differentially expressed genes were screened by gene chip analysis in both HFD and HFD+Oct groups. Color represents the log intensities. Red represents the increased expression genes; green represents the decreased expression genes. The dendrogram demonstrates clustering according to gene classification. The heatmap was plotted using MeV 4.4 software. Oct, octacosanol; HFD, high-fat diet.

**Figure 6 foods-11-01606-f006:**
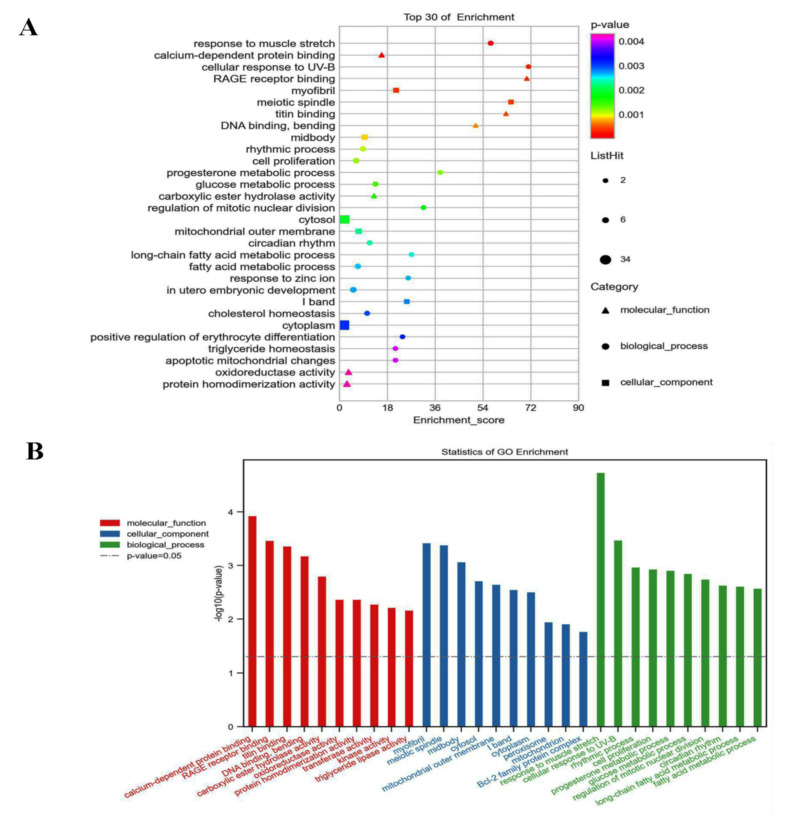
Gene GO analyzed the 72 different expression genes. (**A**) Gene GO analyzed the 72 different expression genes.Bubble chart; (**B**) gene ontology (GO) analysis result. GO analysis consists of biological process, cellular component, and molecular function. The vertical items are the names of gene ontology terms, and the length of horizontal graph represents the adjusted *p*-value. The depth of the color represents the −log(10)-adjusted *p*-value. These analyses using software of g: Profiler and Cytoscape 2.8.2 software.

**Figure 7 foods-11-01606-f007:**
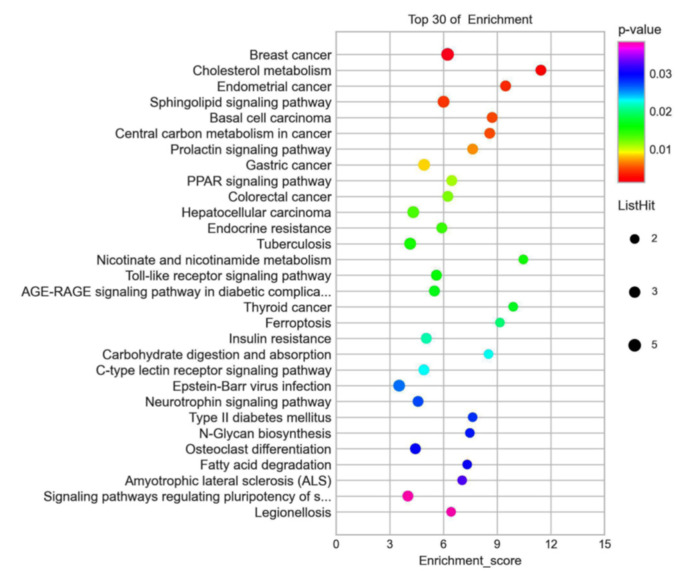
KEGG analyzed the 72 different expression genes. KEGG analysis consists of signal pathways and molecular function. The vertical items are the names of GO terms, and the length of horizontal graph represents the adjusted *p*-value. The depth of the color represents the −log(10) adjusted *p*-value. These analyses using software of g: Profiler and Cytoscape 2.8.2 software.KEGG analyzed the 72 different expression genes.

**Figure 8 foods-11-01606-f008:**
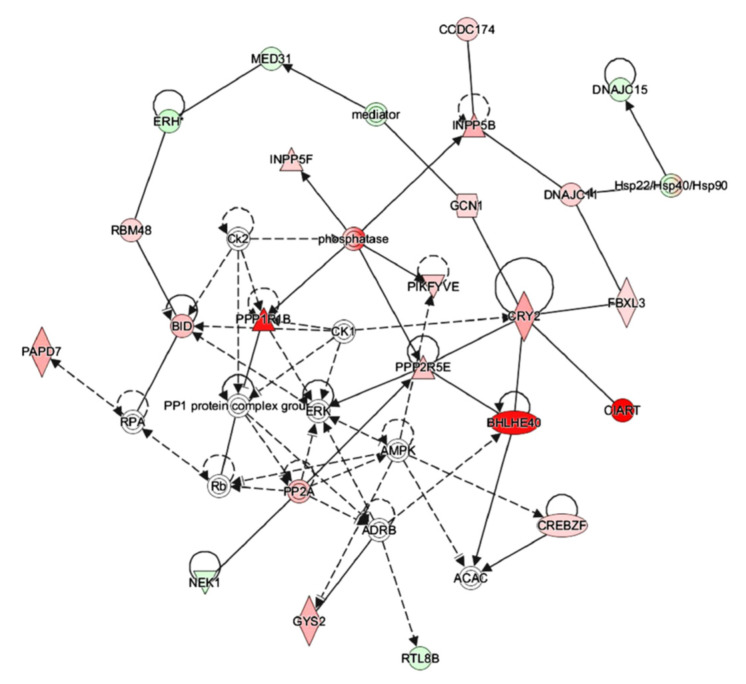
Gene network of octacosanol regulating the lipid metabolism. The normalized gene chip data were uploaded into IPA software to produce the gene network through an automatic comparison between our data with the IPA database. Red, increased expression genes; green, decreased expression genes; and the darkness of the color stands for the ratio of fold change. Gene network of octacosanol regulating the lipid metabolism.

**Figure 9 foods-11-01606-f009:**
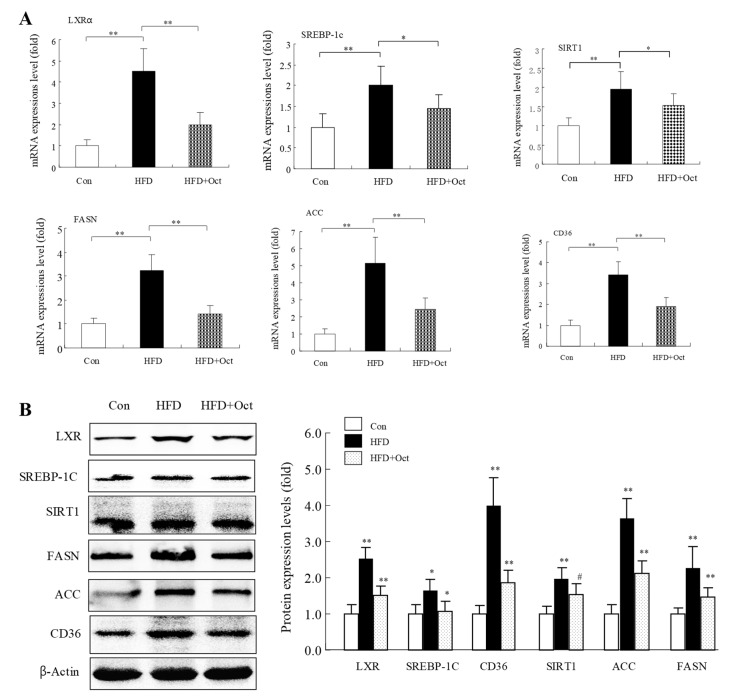
Octacosanol regulated expression levels of lipid metabolism-related genes in liver tissues. (**A**) mRNA expression levels analyzed by RT-PCR (three independent experiments); (**B**) protein expression levels analyzed by WB (three independent experiments). These genes included SREBP-1C, FASN, ACC, SIRT1, and CD36, which were differentially expressed caused by HFD in mouse liver tissue. Oct, octacosanol; HFD, high-fat diet; **: *p* < 0.01; *: *p* < 0.05; ^#^: *p* > 0.05.

**Figure 10 foods-11-01606-f010:**
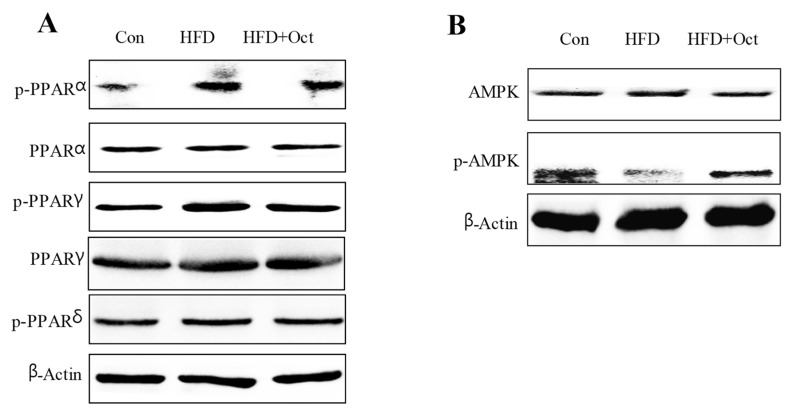
Octacosanol regulated PPAR and AMPK signal pathways in liver tissues. (**A**) Octacosanol regulated the PPAR signal pathway as analyzed by WB (triple duplicate experiments). (**B**) Octacosanol regulated the AMPK signal pathway as analyzed by WB (triple experiments). Oct, octacosanol; HFD, high-fat diet. Octacosanol regulated PPAR and AMPK signal pathways in liver tissues.

**Figure 11 foods-11-01606-f011:**
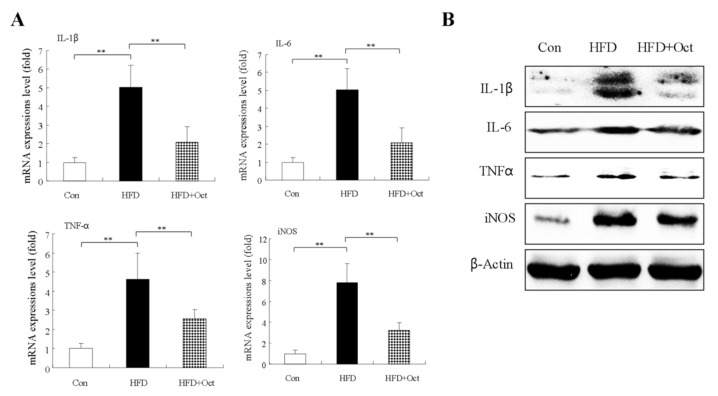
Octacosanol inhibits expression levels of inflammatory factors in the liver tissues of HFD mice. (**A**) Expression levels of cytokines, including IL-6, TNF-α, and iNOS, were examined by the RT-qPCR analysis (three independent experiments). (**B**) WB analyses of the relative expression of IL-6, TNF-α, and iNOS (three independent experiments). Oct, octacosanol; HFD, high-fat diet; **: *p* < 0.01. Octacosanol inhibits expression levels of inflammatory factors in the liver tissues of HFD mice.

**Table 1 foods-11-01606-t001:** Genes upregulated by Oct from gene chip analysis.

ID	Gene Name	HFD/Con Fold Change	HFD/Con *p*-Value	HFD/HFD+Oct Fold Change	HFD/HFD+Oct *p*-Value
ENSMUSG00000026113.17	Inpp4a	0.53	0.036	0.53	0.019
ENSMUSG00000060791.15	Gmfg	0.51	0.014	0.66	0.022
ENSMUSG00000067847.13	Romo1	0.50	0.035	0.75	0.001
ENSMUSG00000060224.3	Pyroxd2	0.47	0.048	0.72	0.028
ENSMUSG00000067653.12	Ankrd23	0.46	0.001	0.57	0.000
ENSMUSG00000044937.14	Ttc41	0.46	0.007	0.60	0.018
ENSMUSG00000025225.14	Nfkb2	0.45	0.022	0.78	0.031
ENSMUSG00000091780.2	Sco2	0.45	0.039	0.59	0.044
ENSMUSG00000003873.11	Bax	0.44	0.014	0.77	0.035
ENSMUSG00000024803.8	Ankrd1	0.44	0.022	0.46	0.008
ENSMUSG00000031845.15	Bco1	0.34	0.021	0.44	0.005
ENSMUSG00000030707.15	Coro1a	0.33	0.044	0.59	0.043
ENSMUSG00000036678.7	Aaas	0.33	0.046	0.63	0.013
ENSMUSG00000006360.11	Crip1	0.30	0.030	0.49	0.015
ENSMUSG00000003555.7	Cyp17a1	0.28	0.033	0.42	0.050
ENSMUSG00000054717.7	Hmgb2	0.26	0.036	0.66	0.018
ENSMUSG00000008496.19	Pou2f2	0.25	0.006	0.42	0.033
ENSMUSG00000037337.11	Map4k1	0.24	0.049	0.49	0.032
ENSMUSG00000022070.6	Bora	0.24	0.028	0.58	0.018
ENSMUSG00000030067.17	Foxp1	0.23	0.032	0.26	0.006
ENSMUSG00000001918.17	Slc1a5	0.22	0.045	0.39	0.031
ENSMUSG00000001020.8	S100a4	0.22	0.043	0.40	0.043
ENSMUSG00000044827.10	Tlr1	0.18	0.044	0.30	0.037
ENSMUSG00000024989.14	Cep55	0.14	0.016	0.36	0.017
ENSMUSG00000078937.8	Cpt1b	0.14	0.022	0.34	0.006
ENSMUSG00000038943.16	Prc1	0.14	0.026	0.58	0.022
ENSMUSG00000028480.14	Glipr2	0.13	0.004	0.21	0.024
ENSMUSG00000045328.11	Cenpe	0.11	0.048	0.41	0.004

**Table 2 foods-11-01606-t002:** Genes down-regulated by Oct from gene chip analysis.

ID	Gene Name	HFD/Con Fold Change	HFD/Con *p*-Value	HFD/HFD+Oct Fold Change	HFD/HFD+Oct *p*-Value
ENSMUSG00000048264.15	Dip2c	2.05	0.026	1.66	0.047
ENSMUSG00000039740.6	Alg2	2.09	0.043	1.36	0.005
ENSMUSG00000030701.17	Plekhb1	2.10	0.035	1.69	0.005
ENSMUSG00000028124.15	Gclm	2.11	0.037	2.21	0.048
ENSMUSG00000068587.9	Mgam	2.11	0.042	1.79	0.005
ENSMUSG00000073565.4	Prr16	2.13	0.028	1.43	0.015
ENSMUSG00000090150.8	Acad11	2.14	0.048	1.60	0.002
ENSMUSG00000035847.15	Ids	2.17	0.034	1.57	0.000
ENSMUSG00000072494.7	Ppp1r3e	2.20	0.010	1.70	0.048
ENSMUSG00000037434.7	Slc30a1	2.25	0.009	1.55	0.010
ENSMUSG00000041237.12	Pklr	2.33	0.039	1.56	0.034
ENSMUSG00000031012.17	Cask	2.43	0.026	1.70	0.015
ENSMUSG00000020102.15	Slc16a7	2.44	0.010	1.79	0.038
ENSMUSG00000047822.7	Angptl8	2.45	0.017	2.04	0.047
ENSMUSG00000027820.12	Mme	2.53	0.005	1.36	0.023
ENSMUSG00000038175.9	Mylip	2.56	0.007	1.24	0.043
ENSMUSG00000024887.9	Asah2	2.57	0.014	2.23	0.029
ENSMUSG00000020262.15	Adarb1	2.59	0.025	2.96	0.012
ENSMUSG00000041685.15	Fcho2	2.72	0.023	2.61	0.023
ENSMUSG00000025059.16	Gk	2.79	0.033	1.60	0.017
ENSMUSG00000020346.16	Mgat1	2.81	0.040	1.73	0.017
ENSMUSG00000053436.14	Mapk14	2.93	0.018	2.59	0.020
ENSMUSG00000014077.13	Chp1	3.00	0.016	1.57	0.027
ENSMUSG00000022887.8	Masp1	3.05	0.031	1.61	0.004
ENSMUSG00000056973.6	Ces1d	3.08	0.002	1.61	0.015
ENSMUSG00000053846.4	Lipg	3.08	0.032	2.05	0.045
ENSMUSG00000024985.18	Tcf7l2	3.12	0.026	2.69	0.014
ENSMUSG00000039662.16	Icmt	3.13	0.013	2.44	0.030
ENSMUSG00000030170.14	Wnt5b	3.14	0.013	1.94	0.042
ENSMUSG00000020572.7	Nampt	3.14	0.004	1.65	0.043
ENSMUSG00000025453.16	Nnt	3.16	0.019	2.69	0.035
ENSMUSG00000033715.14	Akr1c14	3.22	0.020	2.24	0.024
ENSMUSG00000061718.12	Ppp1r1b	3.61	0.048	2.76	0.035
ENSMUSG00000027215.13	Cd82	3.69	0.011	2.33	0.032
ENSMUSG00000018796.13	Acsl1	3.93	0.032	3.48	0.007
ENSMUSG00000030103.11	Bhlhe40	3.99	0.008	2.23	0.028
ENSMUSG00000040127.13	Sdr9c7	4.10	0.009	3.47	0.003
ENSMUSG00000038393.14	Txnip	4.15	0.006	2.33	0.022
ENSMUSG00000037465.9	Klf10	4.27	0.014	1.82	0.000
ENSMUSG00000035349.5	Mia2	4.43	0.043	2.00	0.032
ENSMUSG00000059970.6	Hspa2	4.44	0.015	3.03	0.011
ENSMUSG00000031016.9	Wee1	4.96	0.002	1.47	0.049
ENSMUSG00000032462.14	Pik3cb	6.10	0.008	2.78	0.047
ENSMUSG00000021228.14	Acot3	10.43	0.003	4.28	0.007

## Data Availability

The data presented in this study are available on request from the corresponding author.

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
