# Peer review of "Octacosanol Modifies Obesity, Expression Profile and Inflammation Response of Hepatic Tissues in High-Fat Diet Mice"

_foods, 2022, doi:10.3390/foods11111606_

Round 1

Reviewer 1 Report

It is an interesting study that explores the mechanisms by which a compound, octacosanol (derived from rice grain), modulates the lipid metabolism in the liver of mice fed a high-fat diet (HF-diet). The authors had previously demonstrated that octacosanol reduces fat accumulation and plasma lipid concentration in HF-fed animals. Now, they show that octacosanol modulates different pathways related to lipid metabolism and inflammatory processes. The results are clear and consistent and open the door to the use of this compound as a complement in the therapy for hyperlipidemia.

I have minor points linked to this manuscript that require consideration and revision.

  1. The language is at times confusing, and some sentences are incomprehensible. Please, review grammar throughout the document.

INTRODUCTION

  1. Lines 46-47. Check the sentence, please.
  2. Lines 75-78. Check the sentence, please.

MATERIALS AND METHODS

  1. Section 2.3. The text states that blood glucose concentration is expressed by AUC, but the results shown correspond to blood glucose concentration (Figure 1).
  2. Section 2.4. When the blood is treated with heparin or EDTA clotting is avoided, so what separates after centrifugation is plasma.
  3. Section 2.8. Lines 188-189. The proteins analyzed should be indicated, as well as the antibodies used.
  4. Section 2.9. There is a repeated sentence.

RESULTS

  1. Please check the numbers in the different sections.
  2. Section 3.1. Please review the paragraph. There are sentences that are incorrect (e.g., "the body is almost completely absorbed by the gut").
  3. Figure 1
  • The scale of the y-axis in Figures 1A, 1B and 1C should be the same. If not, it is difficult to see the differences in AUC. It would be interesting to show the results of the AUC, with the explanation of the system used to calculate it. What does the blue line correspond to?
  • The x-axis units must be shown.
  • The legend in the figure should be rewritten. The title of the figure suggests an effect of octacosanol, but does not describe a relevant finding.
  1. Section 3.2. I suggest summarizing this section. It's verbose.
  2. Figure 2.D. Please check. Bars do not correspond to the figures presented in the text (line 241). Letters B, C, and D in the legend are incorrect.
  3. Section 3.4. The paragraph must be rewritten. There are some misleading sentences. Lines 282-287 do not correspond to the results referred to. In addition, the authors should consider differences only in those effects that satisfy the classical P threshold value of less than 0.05. The difference marks in Figure 4.D do not correspond to those indicated in the legend.
  4. It would be interesting to present a scheme of the AMPK and ERK pathways, indicating the key points that the authors have studied (mRNA and protein expression).
  5. Section 2.8. The influence of octacosanol on the PPAR signaling pathway is unclear ("may decrease" "may inhibit" on lines 369-370). The results are presented in the form of Western blot analysis, which is interesting, but the quantification of protein abundance must also be presented.

DISCUSSION

  1. AMPK and SIRT1 are closely related to lipid metabolism and they mutually activate. One would expect to find a regulation of both in the same direction, as discussed in lines 435-443. However, in the results section it is observed that the expression of SIRT1 mRNA is lower in the HFD-Oct group than in the HFD, whereas AMPK expression is modulated in the reverse sense (Figure 9A). Is there any explanation for this discrepancy?
  2. Please check paragraph 459-468. What is said does not correspond to the results presented in Figure 9.

Author Response

Reply to The comments of Reviewer 1

Suggestions for Authors

It is an interesting study that explores the mechanisms by which a compound, octacosanol (derived from rice grain), modulates the lipid metabolism in the liver of mice fed a high-fat diet (HF-diet). The authors had previously demonstrated that octacosanol reduces fat accumulation and plasma lipid concentration in HF-fed animals. Now, they show that octacosanol modulates different pathways related to lipid metabolism and inflammatory processes. The results are clear and consistent and open the door to the use of this compound as a complement in the therapy for hyperlipidemia.

Reply: Thanks you very much for your positive comments.

I have minor points linked to this manuscript that require consideration and revision.

The language is at times confusing, and some sentences are incomprehensible. Please, review grammar throughout the document.

Reply: Thanks your suggestion, Ms Donald is a native English speaker and is studying in our lab, I let him to help us to check the language.

INTRODUCTION

Lines 46-47. Check the sentence, please.

Reply: Thanks, we revised “Hyperlipidemia as a risk factor for cardiovascular disease”

Lines 75-78. Check the sentence, please.

Reply: Thanks. We deleted the sentence “We still do not know what genes and what signal pathways are involved in the effect.” It does not affect to express our idea. Because the last sentence is “But the underlying lipid-lowering mechanism of octacosanol is still unclear”

MATERIALS AND METHODS

Section 2.3. The text states that blood glucose concentration is expressed by AUC, but the results shown correspond to blood glucose concentration (Figure 1).

Reply: Thanks. We revised “Data are described as mean concentration of blood glucose per group using area under the curve”

Section 2.4. When the blood is treated with heparin or EDTA clotting is avoided, so what separates after centrifugation is plasma.

Reply: Thanks very much for good suggestion. We use “plasma” to instead “serum” in the text.

Section 2.8. Lines 188-189. The proteins analyzed should be indicated, as well as the antibodies used.

Reply: Thanks. The pictures just indicated the activation of pathways and they could clearly show that they were activation or inactivation. It is not gene expression. Yes, it is better to do quantitative analysis, but we have 11 images, may be it suitable to put supplementary image. We added used antibodies in the text.

Section 2.9. There is a repeated sentence.

Reply: Thanks. We revised the repeated sentence.

RESULTS

Please check the numbers in the different sections.

Section 3.1. Please review the paragraph. There are sentences that are incorrect (e.g., "the body is almost completely absorbed by the gut").

Reply: Sorry for this type error. It is glucose, we revised the sentence.

Figure 1

The scale of the y-axis in Figures 1A, 1B and 1C should be the same. If not, it is difficult to see the differences in AUC. It would be interesting to show the results of the AUC, with the explanation of the system used to calculate it. What does the blue line correspond to?

Reply: Thanks for you suggestion. I did the image again, and explain the red line and blue line in the image. The calculate method is explained in the text.

The x-axis units must be shown.

Reply: Thanks. x-axis units is labeled with min.

The legend in the figure should be rewritten. The title of the figure suggests an effect of octacosanol, but does not describe a relevant finding.

Reply: Thanks, we checked all legends in the text and revised them.

Section 3.2. I suggest summarizing this section. It's verbose.

Reply: Thanks. We checked this section and revised them.

Figure 2.D. Please check. Bars do not correspond to the figures presented in the text (line 241). Letters B, C, and D in the legend are incorrect.

Reply: Sorry for this. The figure is right and we put spleen weight as fat pad weight. We revised the text and legends.

Section 3.4. The paragraph must be rewritten. There are some misleading sentences. Lines 282-287 do not correspond to the results referred to. In addition, the authors should consider differences only in those effects that satisfy the classical P threshold value of less than 0.05. The difference marks in Figure 4.D do not correspond to those indicated in the legend.

Reply: Thanks. We rewrote the paragraph and revised Figure 4 legends.  

It would be interesting to present a scheme of the AMPK and ERK pathways, indicating the key points that the authors have studied (mRNA and protein expression).

Reply: Thanks you very much for your positive comments.

Section 2.8. The influence of octacosanol on the PPAR signaling pathway is unclear ("may decrease" "may inhibit" on lines 369-370). The results are presented in the form of Western blot analysis, which is interesting, but the quantification of protein abundance must also be presented.

Reply: Thanks. The pictures just indicated the activation of pathways and they could clearly show that they were activation or inactivation. It is not gene expression. Yes, it is better to do quantitative analysis, but we have 11 images, may be it suitable to put supplementary image.  

DISCUSSION

AMPK and SIRT1 are closely related to lipid metabolism and they mutually activate. One would expect to find a regulation of both in the same direction, as discussed in lines 435-443. However, in the results section it is observed that the expression of SIRT1 mRNA is lower in the HFD-Oct group than in the HFD, whereas AMPK expression is modulated in the reverse sense (Figure 9A). Is there any explanation for this discrepancy?

Reply: Thanks! It is interesting and important issue! In fact, although mRNA level of SIRT1 is lower than that in the HFD+Oct group, but the difference between two groups is very small, the p value is just less a little 0.05 (not 0.01). On the contrary, the protein levels of SIRT1 are almost the same in the two groups. We present objective data in the paper. But I doubt SIRT1 is not the important target of octacosanol.

Please check paragraph 459-468. What is said does not correspond to the results presented in Figure 9.

Reply: Thanks. We checked the paragraph and revised the text.

Reviewer 2 Report

The work discloses the mechanism of action of a compound (drug) at various levels including at the gene expression level; 

introduction/objectives: can authors be more clear on whether their research focuses on components of foods/diets or on new drugs/isolated compounds (extracted from plants or chemically synthesised) and why is this work in the scope of the journal and of the current special issue?

authors may wish to consider that rice bran, being a subproduct, would mean that bran being a subproduct, that would mean extracting octacosanol from bran (if viable) could valorise bran and reduce waste? thus contributing to several SDGs? if so, authors are encouraged to elaborate on it and to include supporting references;

line 43:please check possible typo; 

lie 113: please check the coherency and consistency of mM - is that an acronym or a concentration unit?

figure 1: Extra explanations in the figure legend aim at making it more clear to readers; so please complete the legend in order to make the figure   self-explanatory; a) yy axis: explain where is glucose concentration measured and how? b) include units in xx axis in each graph (time?); in the legend please add information about the independent and dependent values and the purpose of the monitoring- ex. evolution of glucose concentration in (….) along time, for a period of 2h (with readings by …. in duplicate/triplicate? each 30 min? c) Authors are suggested to use A: “negative control group”, (followed by the explanation); B: positive control group (followed by the explanation) C: test or intervention group (followed by explanation= intervention), or some other clear distinction of mice groups; please also include n=number of subjects in each group

please avoid "Con", instead, please use C or control (or neg C)...

line 244-fig.2: readjust figures please: A should be presented as a separate figure and the legend should follow the rationale explained in fig.1
the photos can be another figure A and B, keeping the same notation and including a self-explanatory legend
C and D: can be combined into a third figure (same rationale and coherence with the others); 
they can be discussed together but the numbers should be updated along the text; 

line 264-fig.3: please ensure that the most relevant metodological details are summarised in the legend of the figure (don’t forget updating the number)

line 289. fig. 4: please ensure sufficient details are supplied to make the figure self-explanatory;

lines 325. fig. 6: figures should be displayed as independent figures with self-explanatory legends and caption; please use larger and higher resolution pictures, in order to allow reading everything (currently not possible)

line 344. fig. 8: Following the same rationale of self-explanatory figures: please include a brief explanation of the functioning of the algorithm as well as a summarised legend of the symbology, including the designations (or its rationale/grouping) and the different types of arrows and symbols (e.g. circle stands for? triangle stands for?)

line 359. fig.9: As before, please separate in different figure clusters, improve readability and resolution, and include restructure legends and captions to make the figures self-explanatory; don’t forget to update figure numbers; in the case of B, please include the complete photo of the gel aside the current picture where bands are highlighted; do the same in similar situations (as below)

line 392. fig. 11. please separate in 2 different figures and complete information as described before

4. Discussion: authors are encouraged to discuss healthy dietary patterns in preventing disease (e.g. traditional Asian diet as an example - pls see https://www.barillacfn.com/en/publications/a-one-health-approach-to-food/index.html, Lancet commissions etc); In conclusion: if you wish, you may discuss the efficacy of prescribed dietary  regimens in controlling obesity (and linked health issues) with and without drugs (whether “natural” or synthetic); a reflection about improving food literacy of population and or personalised medicine may be worthy to the reader

authors are encouraged to use references from Q1 journals and reference authors (e.g. The Lancet Commissions) in the fields well as including statistics, guidelines and publications from international organisations when appropriate (e.g. WHO, FAO, UN)

The compliance with international ethical standards and requirements concerning the use of animal models, in studies like this one, should be verified. An appropriately detailed statement (and acknowledgements if necessary) should be included, in accordance with the instructions for authors. 

the present work is of quality and brings novelty; with some improvements, it will be even more interesting to the reader. 

Author Response

Reply to the comments of Reviewer 2

The work discloses the mechanism of action of a compound (drug) at various levels including at the gene expression level;

Reply: Thanks very much for your positive comments.

introduction/objectives: can authors be more clear on whether their research focuses on components of foods/diets or on new drugs/isolated compounds (extracted from plants or chemically synthesised) and why is this work in the scope of the journal and of the current special issue?

Reply: We are working in National Research Center of Rice Deep Processing and Byproducts, China. Isolating active compounds from rice bran and investigating their biological functions are our task. Our work will benefit to develop functional foods. For example, we published several papers about oryzanol and we cooperated with Food Company to develop high oryzanol rice bran oil, which has been sold in the Chinese market.

authors may wish to consider that rice bran, being a subproduct, would mean that bran being a subproduct, that would mean extracting octacosanol from bran (if viable) could valorise bran and reduce waste? thus contributing to several SDGs? if so, authors are encouraged to elaborate on it and to include supporting references;

Reply: Octacosanol mainly comes from rice bran, cane and beeswax. All of them are byproducts of food process. Our work is to isolate those compounds and investigate their function and then develop functional foods.

line 43:please check possible typo;

Reply: Thanks, we checked the text and revised them.

lie 113: please check the coherency and consistency of mM - is that an acronym or a concentration unit?

Reply: Thanks, it is concentration unit and appeared a lot of publications.

figure 1: Extra explanations in the figure legend aim at making it more clear to readers; so please complete the legend in order to make the figure   self-explanatory; a) yy axis: explain where is glucose concentration measured and how? b) include units in xx axis in each graph (time?); in the legend please add information about the independent and dependent values and the purpose of the monitoring- ex. evolution of glucose concentration in (….) along time, for a period of 2h (with readings by …. in duplicate/triplicate? each 30 min? c) Authors are suggested to use A: “negative control group”, (followed by the explanation); B: positive control group (followed by the explanation) C: test or intervention group (followed by explanation= intervention), or some other clear distinction of mice groups; please also include n=number of subjects in each group

Reply: Thanks for your suggestion. Sugar tolerance is a very common, classic experiment, I revised the Figure 1, changed time point to experiment time (minutes). We also revised the paragraph and legends. Making the figure self-explanatory is good comments. But now many published papers do not strictly according to this requirement, because the method has been described, adding experimental method in the 11 diagrams note, the paper will be more than the prescribed length, sorry for this.

please avoid "Con", instead, please use C or control (or neg C)...

Reply: Thanks, many other investigators also like to use Con = control. It does not affect understanding of readers.

Experimental design of the 1st part of the study: Con = control, Sil = sildenafil, pax ...

https://figshare.com/articles/figure/_Experimental_design_of_the_1st_part_of_the_study...

mRNA expression of presenilin-1 (PS-1). SED sedentary, CON control, IR... | Download ...

https://www.researchgate.net/figure/mRNA-expression-of-presenilin-1-PS-1-SED-sedentary...

line 244-fig.2: readjust figures please: A should be presented as a separate figure and the legend should follow the rationale explained in fig.1 the photos can be another figure A and B, keeping the same notation and including a self-explanatory legend. C and D: can be combined into a third figure (same rationale and coherence with the others); they can be discussed together but the numbers should be updated along the text;

Reply: Thanks! We revised legend of Figure 2.

line 264-fig.3: please ensure that the most relevant metodological details are summarised in the legend of the figure (don’t forget updating the number)

Reply: Thanks! We revised legend of Figure 3.

line 289. fig. 4: please ensure sufficient details are supplied to make the figure self-explanatory;

Reply: Thanks! We revised legend of Figure 4.  

lines 325. fig. 6: figures should be displayed as independent figures with self-explanatory legends and caption; please use larger and higher resolution pictures, in order to allow reading everything (currently not possible)

Reply: Thanks! We revised. Our original is clear; I do not know why the transformed figure is not clear.

line 344. fig. 8: Following the same rationale of self-explanatory figures: please include a brief explanation of the functioning of the algorithm as well as a summarised legend of the symbology, including the designations (or its rationale/grouping) and the different types of arrows and symbols (e.g. circle stands for? triangle stands for?)

Reply: Thanks ! We revised the legend.

line 359. fig.9: As before, please separate in different figure clusters, improve readability and resolution, and include restructure legends and captions to make the figures self-explanatory; don’t forget to update figure numbers; in the case of B, please include the complete photo of the gel aside the current picture where bands are highlighted; do the same in similar situations (as below)

Reply: Thanks. Our original is clear; I do not know why the transformed figure is not clear.

line 392. fig. 11. please separate in 2 different figures and complete information as described before

Reply: Thanks. Our original is clear; I do not know why the transformed figure is not clear.

  1. Discussion: authors are encouraged to discuss healthy dietary patterns in preventing disease (e.g. traditional Asian diet as an example - pls see https://www.barillacfn.com/en/publications/a-one-health-approach-to-food/index.html, Lancet commissions etc); In conclusion: if you wish, you may discuss the efficacy of prescribed dietary regimens in controlling obesity (and linked health issues) with and without drugs (whether “natural” or synthetic); a reflection about improving food literacy of population and or personalised medicine may be worthy to the reader

Reply: Thanks, we add some content in the discussion part.

authors are encouraged to use references from Q1 journals and reference authors (e.g. The Lancet Commissions) in the fields well as including statistics, guidelines and publications from international organisations when appropriate (e.g. WHO, FAO, UN)

Reply: Thanks! I see.

The compliance with international ethical standards and requirements concerning the use of animal models, in studies like this one, should be verified. An appropriately detailed statement (and acknowledgements if necessary) should be included, in accordance with the instructions for authors.

Reply: Thanks. We described animal ethical in the method part

the present work is of quality and brings novelty; with some improvements, it will be even more interesting to the reader.

Reply: Thanks very much for your positive comment.
